# Application of Autologous Peripheral Blood Mononuclear Cells into the Area of Spinal Cord Injury in a Subacute Period: A Feasibility Study in Pigs

**DOI:** 10.3390/biology10020087

**Published:** 2021-01-24

**Authors:** Iliya Shulman, Sergei Ogurcov, Alexander Kostennikov, Alexander Rogozin, Ekaterina Garanina, Galina Masgutova, Mikhail Sergeev, Albert Rizvanov, Yana Mukhamedshina

**Affiliations:** 1Clinical Research Center for Precision and Regenerative Medicine, Institute of Fundamental Medicine and Biology, Kazan Federal University, 420008 Kazan, Russia; IAShulman@kpfu.ru (I.S.); SVOgurcov@kpfu.ru (S.O.); AleAKostennikov@kpfu.ru (A.K.); AARogozhin@kpfu.ru (A.R.); EEGaranina@kpfu.ru (E.G.); GAMasgutova@kpfu.ru (G.M.); MiASergeev@kpfu.ru (M.S.); Albert.Rizvanov@kpfu.ru (A.R.); 2Republic Clinical Hospital, 420138 Kazan, Russia; 3Department of Neurology, Kazan State Medical Academy–Branch Campus of the Federal State Budgetary Edicational Institution of Father Professional Education «Russian Medical Academy of Continuous Professional Education», 420012 Kazan, Russia; 4Department of Veterinary Surgery, Obstetrics and Small Animal Pathology, Kazan State Academy of Veterinary Medicine, 420029 Kazan, Russia; 5Department of Histology, Cytology, and Embryology, Kazan State Medical University, 420012 Kazan, Russia

**Keywords:** peripheral blood mononuclear cells, spinal cord injury, pigs, fibrin matrix

## Abstract

**Simple Summary:**

Spinal cord injury is a medical and social issue causing severe disability. The potential to overcome the consequences of spinal cord injury is related to cell therapy. Peripheral blood is a prospective and available source of cells for further clinical use. In our study, we have evaluated the therapeutic potential of peripheral blood mononuclear cells (PBMCs) on the model of spinal cord injury in pigs. In the subacute period (6 weeks after injury), PBMCs enclosed in fibrin glue were applied into the dorsal area of the injured spinal cord. In this study, we observed that the tissue integrity increased in the area adjacent to the epicenter of injury, and conduction along spinal axons was partially restored after cell therapy in pigs.

**Abstract:**

Peripheral blood presents an available source of cells for both fundamental research and clinical use. In our study, we have evaluated the therapeutic potential of peripheral blood mononuclear cells (PBMCs) excluding the preliminary sorting or mobilization of peripheral blood stem cells. We have evaluated the regenerative potential of PBMCs embedded into a fibrin matrix (FM) in a model of pig spinal cord injury. The distribution of transplanted PBMCs in the injured spinal cord was evaluated; PBMCs were shown to penetrate into the deep layers of the spinal cord and concentrate mainly in the grey matter. The results of the current study revealed an increase in the tissue integrity in the area adjacent to the epicenter of injury and the partially restored conduction along posterior columns of the spinal cord in animals after FM+PBMC application. The multiplex analysis of blood serum and cerebrospinal fluid showed the cytokine imbalance to occur without significantly shifting toward pro-inflammatory or anti-inflammatory cytokine cascades.

## 1. Introduction

Spinal cord injury (SCI) is both a medical and social issue causing severe disability. Most post-traumatic tissue degeneration is caused by multiple secondary injuries involving several closely related processes such as a blood-brain barrier dysfunction, local inflammation, neuronal death, demyelination, and impaired neural pathways [1]. SCI can lead to a severe motor, sensory, and autonomic dysfunction. At present, there is no effective method to restore an injured spinal cord [2].

There is a possibility to overcome the consequences of SCI by using cell therapy. For this purpose, both stem/progenitor cells and specialized cells (Schwann cells, olfactory epithelial cells, etc.) have been used [2,3,4]. Nevertheless, cells, which might be easily and minimally invasively isolated from the adult body for subsequent autotransplantation to a recipient, remain the most promising for clinical use. Peripheral blood is an attractive and available source for isolation, freezing, and storage for subsequent fundamental research and clinical use.

It has been previously shown that both transplantation of granulocyte colony-stimulating factor-mobilized peripheral blood mononuclear cells (PBMCs) and CD34^+^-PBMCs promoted angiogenesis, axonal regeneration, and the preservation of myelin, contributing to functional recovery [5]. Kijima et al. (2009) demonstrated that human CD133^+^ PBMCs locally transplanted into an injured peripheral nerve of rats could enhance vasculogenesis, providing the microenvironment required for axonal regeneration [6]. The results of the above-mentioned study suggest that mobilized stem fraction of PBMCs might play the role in the creation of a microenvironment promoting neuroregeneration and functional recovery. However, the therapeutic efficacy of PBMCs against SCI, excluding preliminary sorting or mobilization of stem cells, remains unclear.

Scaffolds are used as a vehicle to deliver stem cells into the site of injury; they fill the post-traumatic cavity, thereby promoting axonal growth and acting as a “biological bridge” [7]. The potential of biodegradable natural and synthetic scaffolds has been assessed for maintaining the viability of transplanted cells and creating an enabling microenvironment in vitro and in vivo [7,8,9]. The findings of Itosaka et al. (2009) suggest that a fibrin matrix (FM) might be one of the most promising candidates for a scaffold due to the low risk of developing rejection reactions, it is physiologically more flexible, it has good plasticity, and it is easily implanted into the injured spinal cord [10].

In this study, we aimed to evaluate the therapeutic potential of the application of peripheral blood mononuclear cells embedded in FM in a subacute period of SCI in pigs based on the injured spinal cord structural and functional recovery criteria.

## 2. Materials and Methods

### 2.1. Isolation and Adenoviral Transduction of PBMCs

The study was approved by the Kazan Federal University Animal Care and Use Committee (Permit Number 2, 5 May 2015). Peripheral venous blood was taken from healthy 4-month-old female pot-bellied pigs (8 kg).

Blood samples were collected in vacuum test tubes (Apexlab, Moscow, Russia) supplemented with an ethylenediaminetetraacetic acid (EDTA) solution. All procedures were performed in a biosafety class 2 cell culture laboratory. Primarily, 5 mL of whole blood was mixed with an equal volume of Dulbecco’s phosphate-buffered saline (DPBS, PanEco, Moscow, Russia) in a sterile 15 mL tube. The diluted blood sample was accurately applied to a ficoll solution (ficoll density—1.077 g/cm^3^, PanEco), then centrifuged at 1900 rpm for 20 min without break.

After centrifugation, the fraction of white blood cells was transferred into a new 15 mL tube. The cells were washed with DPBS twice (5 min at 1400 rpm). Red blood cells were lyzed in a lysis solution (155 mM NH_4_Cl, 10 mM KHCO_3_, 0.1 mM EDTA, pH 7.3) for 5 min. The cells were washed with DPBS and resuspended in RPMI-1640 media (PanEco) supplemented with 10% fetal bovine serum (HyClone, Chicago, IL, USA), 2 mM L-glutamine, and 1% mixed penicillin–streptomycin (PanEco). Isolated cells were incubated for 24 h, then washed twice with DPBS.

Isolated PBMCs were seeded onto 10 cm cultural dishes at 8 × 10^6^ cells per plate. The isolated PBMCs were genetically modified with Ad5-EGFP with multiplicity of infection (MOI) 10 and incubated in a humidified chamber containing 5% CO_2_ at 37 °C. Green fluorescence was measured at 24 h by flow cytometry on Guava easyCyte (Merck Millipore). After 24 h, PBMCs transduced with Ad5-EGFP exhibited green fluorescence at a 29.8 ± 4% rate.

### 2.2. Spinal Cord Injury and Experimental Groups

Animals were exposed to surgical manipulations after intubation anesthesia, appropriate pre-operation preparation, and adequate analgesia/pain control. Premedication consisted of intramuscular injection of xylazine (0.6 mg/kg, Bimeda, Dublin, Ireland) and ketamine (5 mg/kg, Hospira, Lake Forest, IL, USA). After propofol induction (IV, 2–6 mg/kg, Fresenius Kabi, Bad Homburg, Germany), endotracheal intubation was conducted using isoflurane (1.3%, Laboratorios Karizoo, Barcelona, Spain) throughout the intervention.

After laminectomy, the anesthetized pigs were subjected to a dosed contusion at the spinal T 10 level with a metal impactor weighing 50 g dropped from a height of 20 cm, followed by compression with the same weight for 10 min. Then, muscles of the back were sutured layer by layer. A urinary catheter (10 Fr, Jorgensen Laboratories Inc., Loveland, CO, USA) was inserted 3–5 days after surgery. Cefazolin (25 mg/kg, Sintez, Kurgan, Russia) and ketoprofen (1 mg/kg, AVZ, Moscow, Russia) were given as intramuscular injections. The pigs were housed separately within the first 48 h, then in pairs.

At 6 weeks after injury, after removing synechiae and making several longitudinal incisions in the dura mater (2–3 incisions with a length of 2–3 mm at a distance of 2–3 mm from each other), 8 × 10^6^ PBMCs per pig, enclosed in 150 μL fibrin glue (Tissucol, Baxter, Deerfield, Illinois, USA) (the experimental group—FM+PBMCs, *n* = 5) were applied on top of the injury. Animals of the control group were applied with cell-free 150 μL FM under similar conditions (the control group—FM, *n* = 5). In addition, two pigs were used to evaluate the distribution of Ad5–EGFP-transduced PBMCs in the area of SCI on day 14 after application. After the wound was sutured layer by layer, cefazoline (25 mg/kg, Pharmasyntez, Moscow, Russia) and ketoprofen (1 mg/kg, Moscow Endocrine Plant, Moscow, Russia) were injected (1 mg/kg) intramuscularly for 5 days.

A subacute period was chosen for cell transplantation as the autologous transplantation of PBMCs is technically feasible in clinical practice when autologous PBMCs derived from a patient can be applied.

### 2.3. Motor Function Testing with PTIBS

To evaluate the effectiveness of motor function recovery, the Porcine Thoracic Injury Behavioral Scale (PTIBS) was used [11]. The PTIBS is a 10-point scale that describes various stages of hindlimb function. Score 1 represents no active hindlimb movement and the rump and knees are on the ground, with Score 10 describing normal ambulation with normal balance. Locomotor recovery in the study groups was video-recorded as previously described [12]. Motor function assessments were scored simultaneously by two supervisors who were blinded to the study groups.

### 2.4. Electrophysiological Studies

Electrophysiological tests were performed for intact and experimental pigs 2 and 11 weeks after SCI as previously described [12]. Figure 1 provides a schematic illustration of the electrode positions on the pigs. The animal’s neuromotor function was assessed by stimulating electromyography. M- and H-waves from the tibialis anterior muscle were recorded in response to stimulation of the sciatic nerve. Monopolar needle electrodes were used for both recording and reference. An active electrode was inserted into the middle of the muscle belly, with the reference electrode implanted within a region of the tendomuscular junction. Electrical stimulation of the sciatic nerve was carried out with square-wave single stimuli lasting for 0.2 ms. For stimulation, monopolar needle electrodes were inserted subcutaneously within the area where the sciatic nerve exits from the pelvis.

To evaluate pyramidal tracts, transcranial electrical stimulation (TES) was used. Motor evoked potentials (MEPs) were registered from the tibialis anterior muscle using the same technique as that for the M-response. Transcranial stimulation was performed by needle electrodes inserted under the scalp up to the contact with the skull bone. The cathode was placed in the middle approximately 0.5 cm caudally from the interorbital line, and the anode was placed in the middle near the occipital bone. Here, 0.1 ms stimuli with intensities ranging from 20 to 400 V were used.

Somatosensory evoked potentials (SEPs) were used to evaluate the posterior columns of the spinal cord. To register them, monopolar needle electrodes were subcutaneously inserted. To register potentials from the lumbar level, an active electrode was inserted over the upper lumbar vertebrae, and the reference electrode was inserted over the middle thoracic vertebrae. For registration of the scalp, an active electrode was inserted over the vertex, and the reference electrode was inserted over the snout. The electrical stimulation of the tail was performed by round electrodes with a stimulus duration of 0.2 ms. The stimulus intensity was chosen by tail movements (the smallest stimulus provoking tail movements was used).

### 2.5. Histological Methods

At 4 months after treatment, the animals were anesthetized and perfused with a 4% paraformaldehyde solution (4 °C). A fragment of the spinal cord (8 cm) was taken from the spinal column and fixed in a 4% paraformaldehyde solution for 2 days. Then, the sample was transferred into 30% sucrose. Cryostat cross-sections of the spinal cord over 1 cm from the injury epicenter rostrally and caudally were stained with azur-eosin. The stained sections were embedded into vitrogel and studied under the APERIO CS2 scanner (Leica). The Aperio imagescope software was used to measure the tissue area.

### 2.6. Cytokine Assay

The multiplex analysis was performed using the MILLIPLEX MAP Porcine Cytokine/Chemokine (magnetic) kit #PCYTMG-23K-13PX (Millipore) as previously described [12]. Each experiment was done in triplicates. The kit enables a simultaneous multiplex analysis of 13 pig cytokines/chemokines/interleukins in a 25-μL aliquot of porcine cerebrospinal fluid (CSF) and blood serum obtained before SCI and on days 7 and 14 post-injury.

### 2.7. Statistical Analysis

To process the results obtained, Origin 7 Pro software was used. Data are presented as mean ± standard deviation (SD). A one-way analysis of variance (ANOVA) with a Tukey’s test were used for multiple comparisons between all study groups. All analyses were performed in a blinded manner with respect to the study groups. The significance level lower than 0.05 (*p* < 0.05) was accepted for all statistical data.

## 3. Results

### 3.1. Distribution of Transplanted PBMCs in Injured Spinal Cord

Fourteen days after the application of Ad5–EGFP-transduced PBMCs, specific fluorescence was detected at least 8 mm rostrally and caudally from the epicenter of injury. EGFP-labeled PBMCs with intense fluorescence clustered in post-traumatic cavities and around them in the epicenter of injury (Figure 2A). At a further distance from the site of the SCI, EGFP^+^-cells were primarily identified in the grey matter and in a small number of ventral and lateral funiculi of the white matter (Figure 2B,C). We also detected EGFP-labeled PBMCs in the area of epidural fibrosis, where more PBMCs were found above the caudal part of the spinal cord, with fewer PBMCs found in the synechias above its rostral part (Figure 2D–G). At the same time, there was a positive correlation caudally, i.e., the greater the distance from the epicenter of the injury, the greater the number of EGFP^+^-cells that were found in the synechias. Rostrally negative correlation was observed, i.e., the further from the site of the SCI, the fewer EGFP^+^-cells in the synechias.

### 3.2. Histological Evaluation of an Injured Spinal Cord after PBMCs Transplantation

We observed the differences in the area of intact tissue and a total area of abnormal cavities between the study groups (Figure 3A–D and Figure 4A–C). The application of FM+PBMCs was associated with an increase in the area of the spared tissue at a distance of 2 mm rostrally and 3 mm caudally from the epicenter of injury (*p* < 0.05) and a decrease in the total area of abnormal cavities at a distance of 3–4 mm caudally from the epicenter of injury (*p* < 0.05) compared to the control group with FM only. Conversely, the morphometry improved in distal areas (6–10 mm rostrally and caudally) from the injury epicenter in the control group treated with FM only.

### 3.3. Changes in Behavioral and Electrophysiology Results

At 6 weeks after the SCI, the pigs had a PTIBS score of 1.64 ± 0.63; active hindlimb movements were either lacking or insignificant (Figure 5A). After the application of FM only or FM+PBMCs, the animals had a similar rise of PTIBS scores within 4 weeks. At 10 weeks after the SCI, the control group (FM alone) had an average PTIBS score of 3.17 ± 0.13 with no further increase in score seen up to 12 weeks. In the experimental group (FM+PBMCs), the PTIBS scores were similar to the control group at 10 weeks and slightly increased after 12 weeks post-injury, reaching a plateau with an average score of 3.74 ± 0.08. However, the differences in PTIBS scores between FM only and FM+PBMCs groups were not statistically significant throughout the study.

Electrophysiological parameters were recorded prior to the injury and at 6 and 22 weeks after the SCI. No significant differences in M-wave amplitude and latency were observed before and after the injury. TES recorded no MEPs and SEPs in all the animals at 6 weeks after the SCI (Figure 5B–D), indicating a profound tissue injury. After application of FM alone or FM+PBMCs, there were no differences in the conduction recovery along the lateral columns: at 22 weeks, only one animal of the control group treated with FM only had MEPs on one side, with MEPs recorded from both limbs in only 1 pig treated with FM+PBMCs. It should be noted that in animals with conduction recovery, the latency of MEPs was lower in those receiving FM+PBMCs. This may suggest a positive effect of PBMCs on the restoration of conduction along spinal axons.

The main electrophysiological differences between the control and the FM+PBMC-treated groups were related to the state of posterior columns of the spinal cord at 22 weeks after the SCI (Figure 5E,F). At 22 weeks, no SEPs from the cortex were recorded in the control group, and only one pig had a lumbar peak. At the same time point, both lumbar and cortical peaks were recorded in two animals of the experimental group (treated with FM+PBMCs), with one animal having a cortical peak without a lumbar one. Thus, the conduction along posterior columns of the spinal cord partially recovered in three out of five pigs applied with FM+PBMCs.

### 3.4. Cytokine Profile

Blood serum and CSF cytokine/chemokine profiles were evaluated on days 7 and 14 after application of FM alone or FM+PBMCs (Figure 6). On day 7, CSF IL-18 level increased and that of IL-1Ra decreased significantly in animals of the experimental group (FM+PBMCs) compared to similar parameters in the control group (FM alone) (Appendix A). In 2 weeks after application, these differences decreased, and the CSF cytokine/chemokine concentrations did not significantly differ in animals of the study groups.

On day 7 after the application of FM+PBMCs, a blood serum IFN-g level significantly increased as compared to the same parameter in the control group (FM alone). Two weeks after the application, IFN-g levels in both groups were similar; however, there were differences in IL-18 concentrations with upregulated levels in the FM+PBMCs group.

## 4. Discussion

The fundamentals of SCI cell therapy have specific features which should be taken into consideration for the regenerative potential of transplanted cells to be most effectively employed. A spinal cord injury is generally an acute condition which requires urgent medical intervention. In this regard, and taking into account the rapid development of inflammatory and immune processes, regenerative therapy with the use of autologous cells in a subacute period of the disorder is preferable [4,13]. At the same time, delayed application of cells is possible as part of a scaffold in a second surgical intervention when a neurosurgeon has access to the surface of the spinal cord. In this case, the peripheral blood presents the most available and convenient source for the clinical use of autologous adult cells.

In our study, PBMCs were for the first time tested in a model of SCI in large animals when evaluating functional and structural parameters as well as blood serum and CSF cytokine/chemokine profiles in a delayed period after therapy. In addition, the migration activity of PBMCs embedded in FM and applied on the area of the SCI was evaluated. The findings indicate that PBMCs can penetrate into the spinal cord and concentrate primarily in the grey matter. However, it should be noted that a part of PBMCs has remained within epidural fibrosis by day 14 after application to the area of the SCI. This might be due to a low PBMC migration rate and the resulting enclosure of some cells in rapidly forming synechias. For example, in our previous study, adipose-derived mesenchymal stem cells embedded in FM transplanted into the area of the SCI in pigs had a higher migration rate into the spinal cord and a smaller number of them in synechias [12]. It is worth noting that PBMCs were not detected in the spinal cord on day 7 after intravenous transplantation to rats with the SCI [14]. When intraspinally transplanted PBSCs were detected 2 mm rostrally and caudally from the epicenter of SCI in mice, their number gradually decreased with time [5].

The behavior analysis with the PTIBS did not show significant differences in the motor function recovery in pigs of the control (FM alone) and experimental (FM+PBMC) groups, despite the positive effect of PBMCs as evidenced by electrophysiological data confirming partial recovery of conduction in the posterior columns of the spinal cord. An intraperitoneal injection of PBMCs-secretome was previously shown in a rat SCI model to improve locomotor activity [15]. Intraspinal transplantation of G–CSF-mobilized peripheral blood stem cells (PBSCs) also promoted functional recovery in a mouse model of SCI [5]. Even a week after a single intravenous injection to rats with SCIs, human peripheral blood-derived CD133^+^ cells provided significant functional recovery [16]. These results in rodents are encouraging; however, it seems to be impossible to compare them with our findings to the full extent as different populations of PBMCs and animal models were used.

That tissue structure can be improved in the setting of PBMC transplantation to rodents with SCIs, as been previously shown; a decreased area of abnormal cavities and less damage of the myelin sheath were noted [5,16,17]. In our study, the application of FM+PBMCs at the site of the SCI improved tissue integrity and reduced the area of abnormal cavities near the epicenter of injury, compared to the control group, without stimulating the regenerative processes at a distance.

The CSF cytokine/chemokine profile showed an increased IL-18 level on day 7 after the application of FM+PBMCs. Being the main immunoregulatory cytokine, IL-18 plays an important role protecting the organism against infections and tumors [18]. At the same time, CSF IL-1Ra level increased in animals treated with FM+PBMCs compared to intact controls (i.e., prior to SCI) and was lower when compared with the SCI group treated with FM alone. Thus, mechanisms preventing the activation of the intracellular inflammatory signaling cascade of IL-1 were noted to develop in both groups, though to a different extent.

The treatment of rat SCI with the PBMCs-secretome was previously found to increase CXCL-1 and IL-10 expression levels in plasma [15]. In our study, serum IL-10 level increased in both study groups compared to intact pigs prior to SCI. However, on day 7 after the application of FM+PBMCs, we detected an upregulated secretion of pro-inflammatory IFN-g in serum samples; IFN-g is known to be one of the triggers of microglia activation and release of neurotoxic factors [19]. At 2 weeks after treatment, IFN-g levels were similar in both groups; however, there were differences in the IL-18 levels, with a higher value in the FM+PBMC group. Thus, our results show the cytokine imbalance to occur without significantly shifting toward pro-inflammatory or anti-inflammatory cytokine cascades. The data obtained confirm our previous research [20], demonstrating the complexity of inflammatory reactions and immunological response after the SCI, as well as their widespread prevalence, which is supported by cytokine profile changes in blood serum. Further investigations of the complex cytokine imbalance that occurs after the SCI, including that associated with cell-based therapy, are required in order to determine whether it can be correlated with the severity of injury and a clinical prognosis.

## 5. Conclusions

Peripheral blood is a convenient and available source of cells which could be feasibly used in clinical practice due to the paracrine mechanism of their action. In our study, we evaluated the therapeutic potential of PBMCs enclosed in FM, excluding preliminary cell sorting or mobilization, when applied in the subacute period of pig contused SCIs. This study would have been improved by the inclusion of a sham operation group, but nonetheless, the results showed the feasibility of the approach and also indicated that tissue integrity was increased in the area adjacent to the epicenter of injury compared to control animals and that conductio along spinal axons was partially recovered in pigs treated with cell therapy.

## Figures and Tables

**Figure 1 biology-10-00087-f001:**
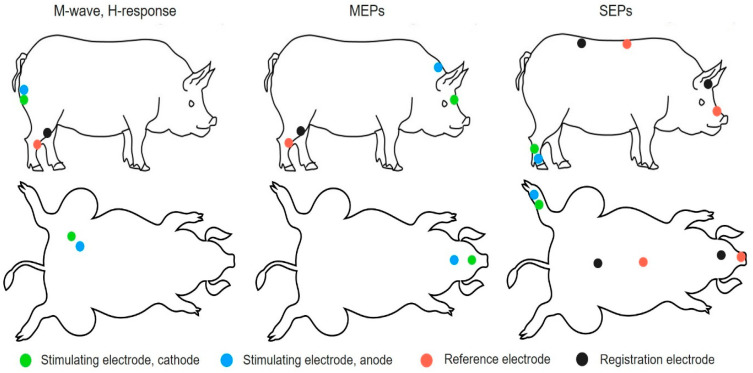
Schematic illustration of electrode positions in pigs.

**Figure 2 biology-10-00087-f002:**
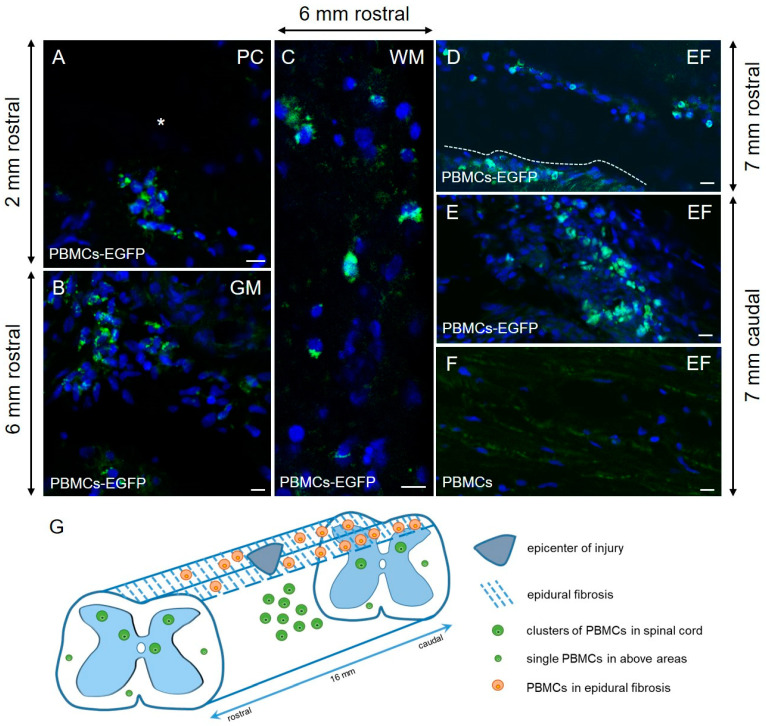
Distribution and survival of peripheral blood mononuclear cells (PBMCs)-EGFP in the area of spinal cord injuries (SCIs) in pigs. On day 14, Ad5–EGFP-labeled (green) PBMCs were located mainly in post-traumatic cavities (PC, asterisk) and around them at the epicenter of trauma (**A**). At a distance from the site of SCIs, EGFP-PBMCs were found mainly in the grey matter ((**B**), GM) and in small amounts in the white matter ((**C**), WM). The EGFP-PBMCs distribution can be seen in the area of epidural fibrosis (**E**); fewer EGFP^+^-cells were found above the rostral part of the spinal cord ((**D**); the dashed line indicates the dorsal surface of the spinal cord), and more in the synechias above its caudal part (**E**). The area of epidural fibrosis in pigs applied with Ad5–EGFP-unlabeled PBMCs is shown in (**F**). Nuclei are stained with 4′,6-diamidino-2-phenylindole (DAPI, blue). Scale bars: 10 µm. Schematic presentation of the EGFP-PBMC distribution in the area of epidural fibrosis and spinal cord tissue according to the distance from the site of injury (**G**).

**Figure 3 biology-10-00087-f003:**
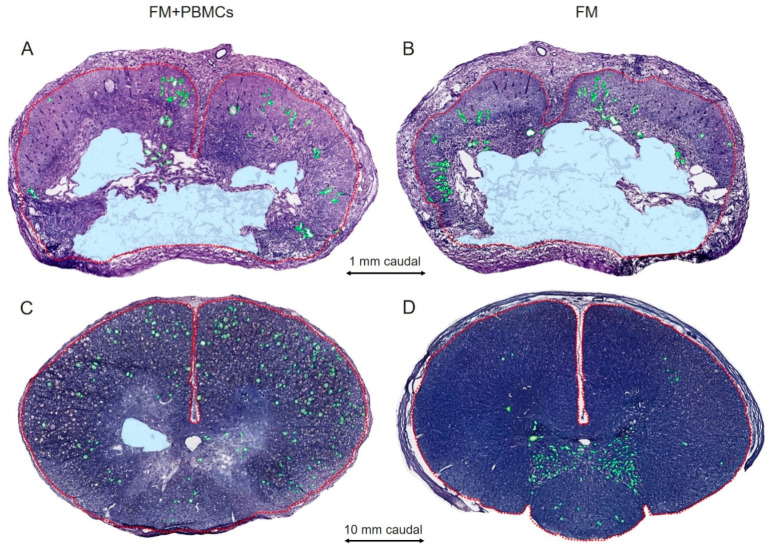
Examples of spinal cord morphometry in experimental pigs. Cross-sections of the injured spinal cord at a distance of 1 (**A**,**B**) and 10 (**C**,**D**) mm caudally from the injury epicenter at 22 weeks after SCIs in experimental pigs. Azur–eosin staining. Dashed red lines indicate the outer border of the spinal cord; green lines and blue areas are examples of small and large post-traumatic cavity determination. Note that areas of the spared tissue in large post-traumatic cavities have also been analyzed.

**Figure 4 biology-10-00087-f004:**
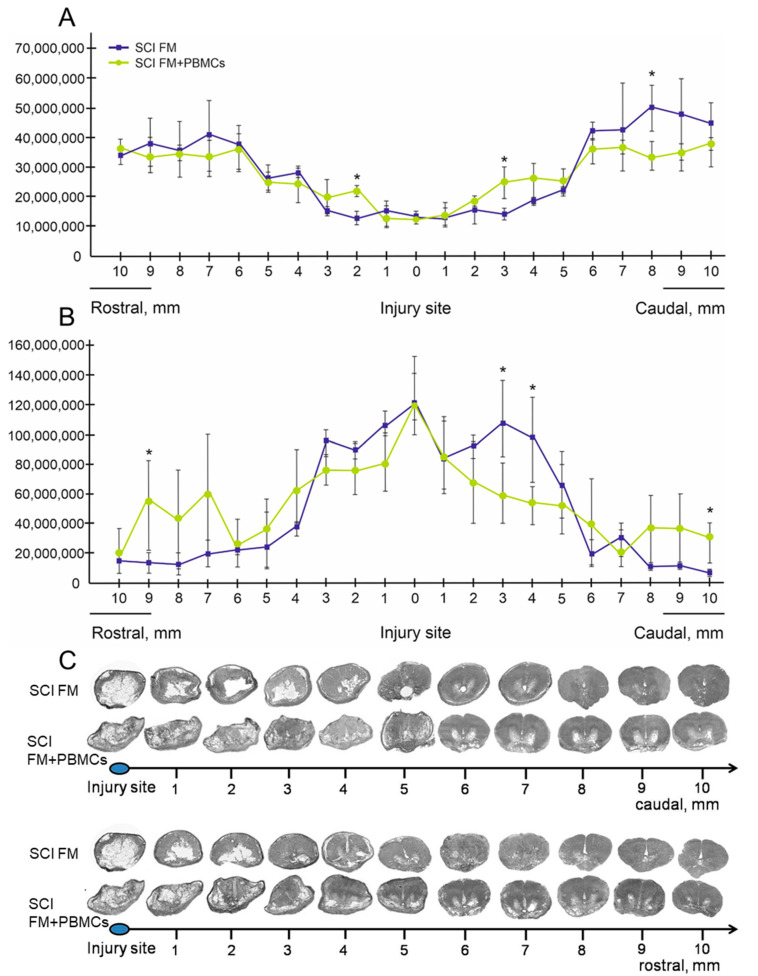
Spinal cord morphometry in experimental pigs. The area of spared tissue (**A**) and the total area of abnormal cavities (**B**) (Y axis, µm^2^)10 mm rostrally and caudally from the injury epicenter 22 weeks after fibrin matrix (FM) (blue line) or FM+PBMC (yellow line) application (5 pigs in each group). * *p* < 0.05, one-way ANOVA followed by a Tukey’s post hoc test. (**C**) Cross-sections of the injured spinal cord at a distance of 10 mm rostrally and caudally from the injury epicenter in 22 weeks after SCIs in experimental pigs. Azur–eosin staining.

**Figure 5 biology-10-00087-f005:**
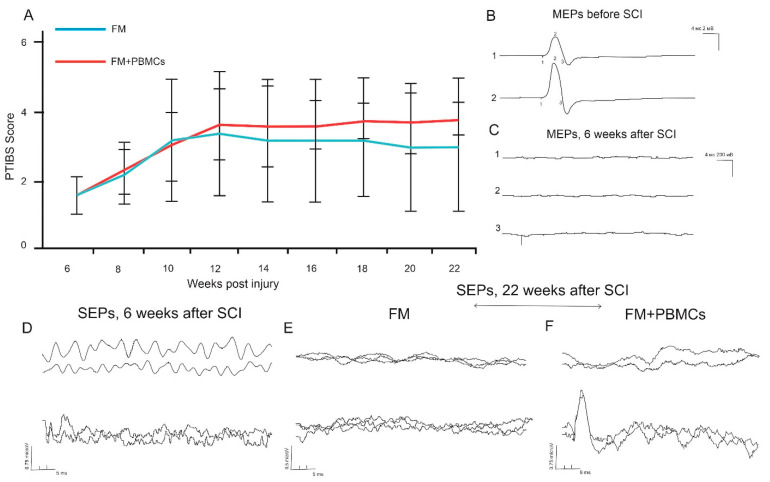
Behavioral testing and electrophysiology results in pigs. Locomotor Porcine Thoracic Injury Behavioral Scale (PTIBS) scores of FM alone (*n* = 5, blue line) and FM+PBMC (*n* = 5, red line) groups (**A**). No significant differences were found between the study groups throughout the study. Electrophysiology results show motor evoked potentials (MEPs) before the SCI (**B**) and MEPs/somatosensory evoked potentials (SEPs) 6 weeks after the SCI (**C**,**D**). After 22 weeks, no cortical or lumbar SEPs were recorded in 4 pigs of the group treated alone after FM (**E**). Both lumbar and cortical peaks (**F**) were recorded in 2 animals of the group treated with FM+PBMCs.

**Figure 6 biology-10-00087-f006:**
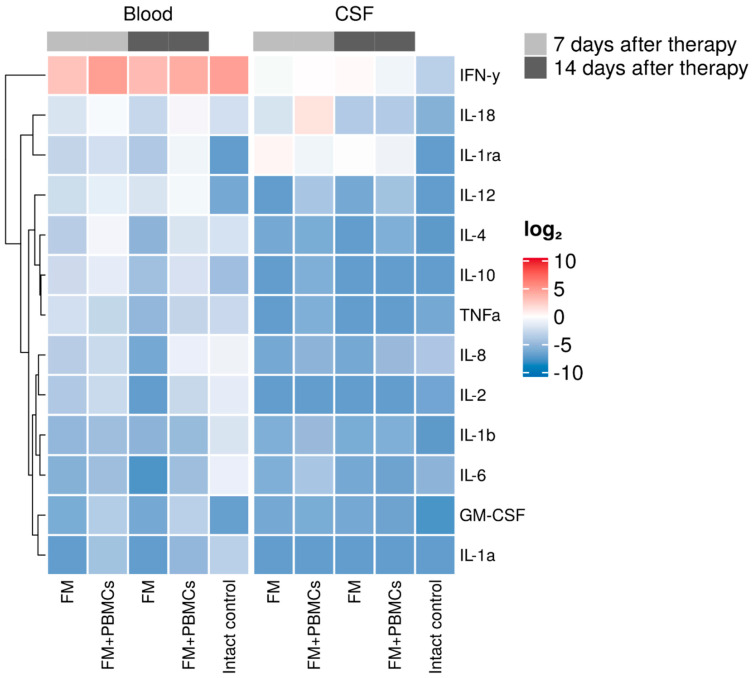
Heat maps showing cytokine/chemokine log10 concentrations, generated with the multiplex analysis of blood serum and cerebrospinal fluid before and after spinal cord injury in the study groups. Cytokine/chemokine profile on days 7 and 14 after FM or FM+PBMCs application (5 pigs in each group). A dendrogram resulting from hierarchical clustering of cytokines is shown on the left.

## Data Availability

The data presented in this study are available on request from the corresponding author. The data are not publicly available due to the evolving nature of the project.

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
