# Peer review of "Application of Autologous Peripheral Blood Mononuclear Cells into the Area of Spinal Cord Injury in a Subacute Period: A Feasibility Study in Pigs"

_biology, 2021, doi:10.3390/biology10020087_

Round 1

Reviewer 1 Report

I think that this re-revised paper entitled: Application of autologous peripheral blood mononuclear cells into the area of spinal cord injury in a subacute period: a pilot study in pigs, is now satisfactory improved and ready to the publication.

Author Response

We would like to thank the reviewer for this review and approval the publication of this paper.

Reviewer 2 Report

The authors have made corrections to the orginally submitted manuscript. However, many of my suggestions are not adequately addressed.

Most importantly, power calculations must be provided. Secondly, in all instances, "Pilot study" MUST be replaced by "Feasibility study".

Please also see my original peer review. 

Author Response

The authors have made corrections to the orginally submitted manuscript. However, many of my suggestions are not adequately addressed.

Most importantly, power calculations must be provided. Secondly, in all instances, "Pilot study" MUST be replaced by "Feasibility study".

Please also see my original peer review. 

Authors: According to the reviewer’s comment, we have replaced "Pilot study" by "Feasibility study". We have also revised all Manuscript (see our revision, using the "Track Changes" function). According to the Editor’s comment, the limitation regarding sham controls was replaced in the Conclusion.

This manuscript is a resubmission of an earlier submission. The following is a list of the peer review reports and author responses from that submission.

Round 1

Reviewer 1 Report

This study described the application of autologous peripheral blood mononuclear cells in the area of spinal cord injury. This is the large animal experiment, thus, interesting. However, several unclear and/or insufficient points are pointed out, and there are significantly reducing the quality of this study and should be improved.     

Comments:

  1. Spell-out DPBS.
  2. Actually, PBMCs (peripheral blood mononuclear cells) are rude and confusing abbreviated name. More correctly, there are the peripheral blood mononuclear cell-derived multipotent progenitor cells, isn’t it?
  3. Similarly, dose the MSCs means what kind of MSCs, bone marrow derived stromal cells? Clarify these points is kinder to the readers.
  4. Although, the recording of electrophysiological results is fine, but the method of electrophysiological assessment is unclear. The authors reported that “Electrophysiological tests were performed for intact and experimental pigs 2 and 11 weeks after SCI as previously described [15]”. However, there was no information in the ref 15, this is the rat experiment with the BBB open field locomotor test. Clarify the relationship among stimulation of the sciatic nerve, M- and H-waves, pyramidal tracts transcranial electrical stimulation and M-response etc. Appropriate schematic drawing should be added. This may be a good contribution to the establishment of the method for the large animal electrophysiology.
  5. Did the PBMCs which was used in this study was CD45 positive? How was the CD133?  How was the transfection rate of EGFP? I think this information is the minimum requirement of this kind of study.
  6. In the Figure 1E, there are many GFP+ reactions without link to nuclear (DAPI negative). What is it? Platelet?
  7. Similarly, what is the difference between PBMCs and single PBMCs in panel G.
  8. Does the PBMCs in spinal cord is not single cell?
  9. I cannot image the panel G scheme from the photograph in A to F. Low power shot and large photo, which could be connected with A-F to G should be added (such as may be large photo of Fig. 2C with dark field).
  10. 2C: cross-sectional photos are too small to identify normal and abnormal area. Pick up several specific photographs and enlarged.
  11. Table 1 is also complicated and almost indistinguishable. Reduced the digit number, may be ably using cubes and/or quadruplicate manner.

Author Response

This study described the application of autologous peripheral blood mononuclear cells in the area of spinal cord injury. This is the large animal experiment, thus, interesting. However, several unclear and/or insufficient points are pointed out, and there are significantly reducing the quality of this study and should be improved.

Authors: We would like to thank the reviewer for this review and pointing out some errors. We hope that our next responses will satisfy the reviewer. The manuscript has also been corrected by a proficient English speaker.

Comments:

  1. Spell-out DPBS.

Authors: Done.

  1. Actually, PBMCs (peripheral blood mononuclear cells) are rude and confusing abbreviated name. More correctly, there are the peripheral blood mononuclear cell-derived multipotent progenitor cells, isn’t it?

Authors: Unfortunately, it is not. We isolated mononuclear fraction of peripheral blood cells without subsequent sorting of cells or preliminary stimulation of stem cell release from the bone marrow. Immunophenotype obtained PBMCs presented in the following answers.

  1. Similarly, dose the MSCs means what kind of MSCs, bone marrow derived stromal cells? Clarify these points is kinder to the readers.

Authors: There are 3 mentions about MSCs in our article. In every case we clarified kind of MSCs.

  1. Although, the recording of electrophysiological results is fine, but the method of electrophysiological assessment is unclear. The authors reported that “Electrophysiological tests were performed for intact and experimental pigs 2 and 11 weeks after SCI as previously described [15]”. However, there was no information in the ref 15, this is the rat experiment with the BBB open field locomotor test. Clarify the relationship among stimulation of the sciatic nerve, M- and H-waves, pyramidal tracts transcranial electrical stimulation and M-response etc. Appropriate schematic drawing should be added. This may be a good contribution to the establishment of the method for the large animal electrophysiology.

Authors: We apologize for our mistake. All ref. has been verified. We have added additional figure (fig.1 in new version) illustrating the method of large animal electrophysiology in Material and methods.

  1. Did the PBMCs which was used in this study was CD45 positive? How was the CD133?  How was the transfection rate of EGFP? I think this information is the minimum requirement of this kind of study.

Authors: We performed flow cytometry of PBMCs with the use of antibodies against CD3 (84±17,4%), CD4 (72,7±24,7%), CD8 (37,5±23,3%), CD14 (91,3±6,2%), CD45 (92,6±3,9%) and CD34 (5,6±7%). Unfortunately, we did not analyze expression of CD133 in PBMCs. According to the reviewer’s comment, we have added information about transfection rate of EGFP.

  1. In the Figure 1E, there are many GFP+ reactions without link to nuclear (DAPI negative). What is it? Platelet?

Authors: We agree with the reviewer’s comment that GFP+ reactions without link to nuclear in large quantities, at least it seems strange. The first thought arises about non-specific reaction or autofluorescence. However, we did not observe GFP+ reactions in histological sections of spinal cord in pigs without EGFP-labeled PBMCs application (Fig.2F in new version). Number of platelets in obtained PBMCs was very low. Therefore, this phenomenon cannot be explained by this either. Since at the moment we cannot give an exact answer to this question, we decided to replace the fig.2E with a similar photo of another pig with EGFP-labeled PBMCs application.

  1. Similarly, what is the difference between PBMCs and single PBMCs in panel G.

Authors: According to the reviewer’s comment, we have added clarification on this matter in Fig. 1G.

  1. Does the PBMCs in spinal cord is not single cell?

Authors: We observed single GFP+ PBMCs predominantly in white matter away from the epicenter of injury. Nonetheless, GFP+ PBMCs were mainly visualized as cell clusters in gray matter and epicenter of injury.

  1. I cannot image the panel G scheme from the photograph in A to F. Low power shot and large photo, which could be connected with A-F to G should be added (such as may be large photo of Fig. 2C with dark field).

Authors: If we understand the reviewer, we could have added large photo of total section of spinal cord for identification of complete GFP+ PBMCs distribution. We agree with the reviewer’s comment that such additional photo could further strengthen the data presented. However, our experience shows that visualization of total section of spinal cord is not as presentable as we would like. GFP expression is single, distributed in different areas of the spinal cord at a distance from the site of SCI. But at the epicenter of SCI there are many nonspecific reactions (visualized in all channels by confocal microscope) in large post-traumatic cavities. In this regard, we considered that a variant of schematic presentation of the EGFP-PBMCs distribution is the most optimal.

  1. 2C: cross-sectional photos are too small to identify normal and abnormal area. Pick up several specific photographs and enlarged.

Authors: We agree with the reviewer’s comment and have added new Figure (Fig. 3 in new version) with specific enlarged photographs of cross-sectional spinal cord.

  1. Table 1 is also complicated and almost indistinguishable. Reduced the digit number, may be ably using cubes and/or quadruplicate manner.

Authors: Thank you for your remarks. I agree, that the table looks complicated. However due to the huge differences in cytokine concentrations using cubes and/or quadruplicate manner might be not effective. We think that it would be better to input this table in supplementary material. In addition, figure 6 demonstrates well the differences in cytokine expression pattern.

Reviewer 2 Report

In their manuscript, Shulman et al. applied autologous PBMCs embedded within a fibrin matrix in a pig model of spinal cord injury. Briefly, GFP-labelled pig PBMCs embedded within fibrin matrices were transplanted into pigs 6 weeks post injury followed by assessment of the motor function, electrophysiological recording of M- and H- waves, histology and analysis of the cytokine profile in the blood and in the cerebrospinal fluid.  Although not very exciting, the study is still interesting. However, several flaws need to be addressed.

Major issues:

  1. SHAM controls (no FM, no PBMCs) are missing. In an ideal scenario, this should be included. If the authors do not have the capacity to provide this information, they should at lead make this limitation very clear throughout the manuscript.
  2. The n-numbers for the individual groups do not match (only 2 animals for d14). The authors should provide power calculations for the design of the in vivo experiments.
  3. Materials and methods: batch number of the FCS needs to be provided.
  4. How long was the incubation period prior to transplantation?
  5. How many cells have been transplanted? What was the rationale for using these cell numbers?
  6. How exactly did the authors perform the transplantations?

Minor issues:

Page 2, line 48: “stem and progenitor cells, which can be easily and invasively isolated from an adult body for subsequent autotransplantation to a recipient are most promising for clinical use.”

This statement (a the subsequent sentence) is overly biased towards haematopoietic stem cells. Although this is indeed one of potential stem cell types that can be applied, purified MSCS are the most widely used stem cell type. The authors should soften this statement.

Page 2, line 56: “Several populations of progenitor cells isolated from PBMCs can also differentiate into neurons and 56 glia [5–7].”

This claim is nonsense and has been proven wrong on multiple occasions. Also, there is not a single serious publication showing that PBMCs can cross the germ layer boundary. I the authors disagree, they should show this in experiments and include electrophysiological comparison with mature neurons.  It is nowadays a scientific consensus that non-neural adult stem cells participate to neural regeneration via paracrine bystander effects (extracellular vesicles) and not via direct integration and differentiation. This needs to be included in the introduction and in the discussion.

Author Response

In their manuscript, Shulman et al. applied autologous PBMCs embedded within a fibrin matrix in a pig model of spinal cord injury. Briefly, GFP-labelled pig PBMCs embedded within fibrin matrices were transplanted into pigs 6 weeks post injury followed by assessment of the motor function, electrophysiological recording of M- and H- waves, histology and analysis of the cytokine profile in the blood and in the cerebrospinal fluid.  Although not very exciting, the study is still interesting. However, several flaws need to be addressed.

Authors: We would like to thank the reviewer for this review and pointing out some errors. We hope that our next responses will satisfy the reviewer. The manuscript has also been corrected by a proficient English speaker.

Major issues:

  1. SHAM controls (no FM, no PBMCs) are missing. In an ideal scenario, this should be included. If the authors do not have the capacity to provide this information, they should at lead make this limitation very clear throughout the manuscript.

Authors: We agree with the reviewer’s comment that Sham control group could be include in an ideal experimental design. Nonetheless, we limited in quantity of experimental pig according to decision Animal Care and Use Committee. According to the reviewer’s comment, we have added this limitation very clear throughout the manuscript.

  1. The n-numbers for the individual groups do not match (only 2 animals for d14). The authors should provide power calculations for the design of the in vivo experiments.

Authors: Our study was a pilot investigation using big animal model of spinal cord injury. In this regard we were limited in quantity of experimental pig according to the decision of the Animal Care and Use Committee. There are two tested groups in our study: FM + PBMCs and FM with n=5 pigs in each groups. We also added 2 additional pigs for evaluation «the distribution of LV-EGFP-transduced PBMCs in the area of SCI at day 14 after application». In this case, Animal Care and Use Committee thought that 2 pigs were enough for assessment of cells distribution already. So, our pilot study included 12 pigs in experimental investigation.

  1. Materials and methods: batch number of the FCS needs to be provided.

Authors: Could you please specify what do you mean under the term FCS?. Unfortunately, this abbreviation is not utilized in manuscript.

  1. How long was the incubation period prior to transplantation?

Authors: We have added this information in Materials and methods.

  1. How many cells have been transplanted? What was the rationale for using these cell numbers?

Authors: 8 × 106 PBMCs per pig, enclosed in 150 μL fibrin glue were applied on top of the injury. Number of transplanted cells corresponds to the dose 1 million cells/kg of body weight in pigs. The amount of fibrin glue is also not taken by chance: 150 μL is necessary so that the fibrin glue lies flat on the surface of the spinal cord, without flowing beyond borders of laminectomy.

  1. How exactly did the authors perform the transplantations?

Authors: According to the reviewer’s comment, we have added exact information about transplantation in Material and methods.

Minor issues:

Page 2, line 48: “stem and progenitor cells, which can be easily and invasively isolated from an adult body for subsequent autotransplantation to a recipient are most promising for clinical use.” This statement (a the subsequent sentence) is overly biased towards haematopoietic stem cells. Although this is indeed one of potential stem cell types that can be applied, purified MSCS are the most widely used stem cell type. The authors should soften this statement.

Authors: We agree with the reviewer’s comment and have changed above statement: « Nevertheless, stem and progenitor cells, which can be easily and minimally invasively isolated from an adult body for subsequent autotransplantation to a recipient are most promising for clinical use. Peripheral blood is a convenient but not the only source from which patients' stem and progenitor cells can be isolated, frozen and stored for subsequent use. ».

Page 2, line 56: “Several populations of progenitor cells isolated from PBMCs can also differentiate into neurons and 56 glia [5–7].” This claim is nonsense and has been proven wrong on multiple occasions. Also, there is not a single serious publication showing that PBMCs can cross the germ layer boundary. I the authors disagree, they should show this in experiments and include electrophysiological comparison with mature neurons.  It is nowadays a scientific consensus that non-neural adult stem cells participate to neural regeneration via paracrine bystander effects (extracellular vesicles) and not via direct integration and differentiation. This needs to be included in the introduction and in the discussion.

Authors: We agree with the reviewer’s comment and have added next sentence: «Despite the fact that earlier publications claimed the possibility of progenitor cells isolated from PBMCs could differentiate into nerve cells [5–7], this has not been proven to date. The therapeutic effect of non-neural adult stem and progenitor cells is due to a paracrine mechanism of their action. »

Round 2

Reviewer 1 Report

According to the several changes and the addition of several Figures, this paper getting better. However, two big matters are raised from the hearing of authors' responses.

  1. Authors: Unfortunately, it is not. We isolated mononuclear fraction of peripheral blood cells without subsequent sorting of cells or preliminary stimulation of stem cell release from the bone marrow. Immunophenotype obtained PBMCs presented in the following answers.
  2. Authors: There are 3 mentions about MSCs in our article. In every case we clarified kind of MSCs.
  3. Authors: We performed flow cytometry of PBMCs with the use of antibodies against CD3 (84±17,4%), CD4 (72,7±24,7%), CD8 (37,5±23,3%), CD14 (91,3±6,2%), CD45 (92,6±3,9%) and CD34 (5,6±7%). Unfortunately, we did not analyze expression of CD133 in PBMCs. (By the way, this information should be added as the characteristics of PBMCs)

From the authors' response of the above, it was clear that the PBMCs did not show any of stem cell characteristics. Thus, I estimated that there are wholly hematopoietic cells without the red blood cells. In this regard, description of stem and/or progenitor cells or MSCs, as a common denominator of the present PBMCs, is actually inappropriate.  

In relation to the above, I suppose that a single EGFP+ reactions and/or EGFP cluster in the gray matter without nuclear may be a debris of phagocytosis. If this is the fact, the results of this experiment wholly induced inflammatory response derived factors.    

These two issues should be explained in the text clearly. These are fundamental critical points of this study, thus, should be included at this time before the publication.  

Specify the use of Adenoviral vector in the method, because of the addition of the very high transfection rate (over 90%) of EGFP even for the non-proliferating cell population.

Author Response

According to the several changes and the addition of several Figures, this paper getting better. However, two big matters are raised from the hearing of authors' responses.

  1. Authors: Unfortunately, it is not. We isolated mononuclear fraction of peripheral blood cells without subsequent sorting of cells or preliminary stimulation of stem cell release from the bone marrow. Immunophenotype obtained PBMCs presented in the following answers.
  2. Authors: There are 3 mentions about MSCs in our article. In every case we clarified kind of MSCs.
  3. Authors: We performed flow cytometry of PBMCs with the use of antibodies against CD3 (84±17,4%), CD4 (72,7±24,7%), CD8 (37,5±23,3%), CD14 (91,3±6,2%), CD45 (92,6±3,9%) and CD34 (5,6±7%). Unfortunately, we did not analyze expression of CD133 in PBMCs. (By the way, this information should be added as the characteristics of PBMCs)

From the authors' response of the above, it was clear that the PBMCs did not show any of stem cell characteristics. Thus, I estimated that there are wholly hematopoietic cells without the red blood cells. In this regard, description of stem and/or progenitor cells or MSCs, as a common denominator of the present PBMCs, is actually inappropriate.  

Authors: We agree with the reviewer’s comment and rewrote relevant sentences.

In relation to the above, I suppose that a single EGFP+ reactions and/or EGFP cluster in the gray matter without nuclear may be a debris of phagocytosis. If this is the fact, the results of this experiment wholly induced inflammatory response derived factors.    

These two issues should be explained in the text clearly. These are fundamental critical points of this study, thus, should be included at this time before the publication.  

 Authors: The speculation of phagocytosis of transplanted EGFP+ cells takes place, however, requires convincing evidence and additional research not provided in this manuscript. Regarding induction of inflammatory response, the multiplex analysis of blood serum and cerebrospinal fluid did not show severe displacement of the cytokine level towards pro-inflammatory responses.

Specify the use of Adenoviral vector in the method, because of the addition of the very high transfection rate (over 90%) of EGFP even for the non-proliferating cell population.

Authors: We apologize for our mistake, transduction efficiency of PBMCs with Ad5-EGFP was 29.8±4% according to flow cytometry results. Unfortunately, we previously erroneously presented another data concerning another transduction approach.

P.S. The manuscript has been corrected again by a proficient English speaker.

Reviewer 2 Report

  1. In the revised version of their manuscript Shulman et al have addressed some of my previous concerns. Although the author`s answer to my concern regarding the lack of sham controls and insufficient numbers of subjects have been clearly mentioned as a limitation, this is still a major flaw of the study. A potential solution to the lack of statistical power and controls would be a relabelling the study from pilot study to feasibility study. The authors should change all respective references and soften all statements that would indicate changes (e.g. improvement). Examples of such statements are listed below:

“ This pilot study demonstrated that the tissue structure could be improved and the conduction along spinal axons could be partially recovered in pigs using cell therapy.”

“ The current pilot study demonstrated that the tissue structure could be improved and the conduction along posterior columns of the spinal cord could be partially recovered in animals using FM+РВМCs application”

“We have studied the therapeutic potential”

“In our study we have evaluated the therapeutic potential of PBMCs, enclosed in FM, application in subacute period of pig contused SCI. This pilot study demonstrated that the tissue structure could be improved and the conduction along spinal axons could be partially recovered in pigs with cell therapy.”

  1. FCS = fetal calf serum / fetal bovine serum. Batch/lot number is still missing
  2. Figure 4. number of animals per group needs to indicated
  3. Figure 5 number of animals per group needs to indicated
  4. Figure 6 number of animals per group needs to indicated
  5. Transduction of cells can change the cells. Missing control (empty vector) needs to be mentioned as limitation,

Author Response

Reviewer 2

  1. In the revised version of their manuscript Shulman et al have addressed some of my previous concerns. Although the author`s answer to my concern regarding the lack of sham controls and insufficient numbers of subjects have been clearly mentioned as a limitation, this is still a major flaw of the study. A potential solution to the lack of statistical power and controls would be a relabelling the study from pilot study to feasibility study. The authors should change all respective references and soften all statements that would indicate changes (e.g. improvement). Examples of such statements are listed below:

 “ This pilot study demonstrated that the tissue structure could be improved and the conduction along spinal axons could be partially recovered in pigs using cell therapy.”

“ The current pilot study demonstrated that the tissue structure could be improved and the conduction along posterior columns of the spinal cord could be partially recovered in animals using FM+РВМCs application”

“We have studied the therapeutic potential”

“In our study we have evaluated the therapeutic potential of PBMCs, enclosed in FM, application in subacute period of pig contused SCI. This pilot study demonstrated that the tissue structure could be improved and the conduction along spinal axons could be partially recovered in pigs with cell therapy.”

Authors: In our previous reply we have forgotten to mention that fibrin glue (Tissucol, Baxter) applied in the control group is known to be high effective local hemostatic medicine, frequently used in modern neurosurgery and in case of dura matter damage. It has been previously shown that fibrin glue (Baxter) has been used in many large clinical series without adverse events (doi: 10.1016/j.spinee.2010.09.017). Thus including additional control group without application of fibrin glue to the spinal cord surface remains dispute.

Taking in account above mentioned, we suggest that title of the article reflects the idea of investigation. However, following reviewer`s recommendations we included the following sentence to the Materials and Methods section «5 animals in each group were feasible sample size for our research group».

According to reviewer comment, we have also soften all statements that indicate changes/improvement.

  1. FCS = fetal calf serum / fetal bovine serum. Batch/lot number is still missing

Authors: Done.

  1. Figure 4. number of animals per group needs to indicated

Figure 5 number of animals per group needs to indicated

Figure 6 number of animals per group needs to indicated

Authors: Done in fig. 4, 5, 6 legends.

  1. Transduction of cells can change the cells. Missing control (empty vector) needs to be mentioned as limitation

Authors: Our preliminary studies have shown that no effect at the level of the cell transcriptome is observed in the case of cell transduction by Ad5-EGFP (doi.org/10.1182/blood-2020-138664).

P.S. The manuscript has been corrected again by a proficient English speaker.